# Neurotrophin Signaling Impairment by Viral Infections in the Central Nervous System

**DOI:** 10.3390/ijms23105817

**Published:** 2022-05-22

**Authors:** Karen Bohmwald, Catalina A. Andrade, Valentina P. Mora, José T. Muñoz, Robinson Ramírez, María F. Rojas, Alexis M. Kalergis

**Affiliations:** 1Millennium Institute on Immunology and Immunotherapy, Departamento de Genética Molecular y Microbiología, Facultad de Ciencias Biológicas, Pontificia Universidad Católica de Chile, Santiago 8331150, Chile; kbohmwald@uc.cl (K.B.); cnandrade@uc.cl (C.A.A.); valentina.mora@uc.cl (V.P.M.); jos.muoz@uc.cl (J.T.M.); robinson.r@uc.cl (R.R.); marferh@uc.cl (M.F.R.); 2Departamento de Endocrinología, Facultad de Medicina, Pontificia Universidad Católica de Chile, Santiago 8331150, Chile

**Keywords:** neurotrophins, neurotrophins signaling, viral infection, neurological alterations

## Abstract

Neurotrophins, such as nerve growth factor (NGF), brain-derived neurotrophic factor (BDNF), and neurotrophin 3 (NT-3), NT-4, and NT-5, are proteins involved in several important functions of the central nervous system. The activation of the signaling pathways of these neurotrophins, or even by their immature form, pro-neurotrophins, starts with their recognition by cellular receptors, such as tropomyosin receptor kinase (Trk) and 75 kD NT receptors (p75NTR). The Trk receptor is considered to have a high affinity for attachment to specific neurotrophins, while the p75NTR receptor has less affinity for attachment with neurotrophins. The correct functioning of these signaling pathways contributes to proper brain development, neuronal survival, and synaptic plasticity. Unbalanced levels of neurotrophins and pro-neurotrophins have been associated with neurological disorders, illustrating the importance of these molecules in the central nervous system. Furthermore, reports have indicated that viruses can alter the normal levels of neurotrophins by interfering with their signaling pathways. This work discusses the importance of neurotrophins in the central nervous system, their signaling pathways, and how viruses can affect them.

## 1. Introduction

Neurotrophins (NTs), also known as neurotrophic factors, are proteins involved in many important functions in the central nervous system (CNS), such as survival, synaptic plasticity, apoptosis, differentiation, and growth [1,2,3]. The current knowledge of NTs includes the nerve growth factor (NGF), neurotrophin 3 (NT-3), NT-4, NT-5, and brain-derived neurotrophic factor (BDNF). These NTs are commonly known to be synthesized by CNS cells [4,5,6]. However, other cell types, including immune system cells, can produce NTs and express their receptors on the cell surface [7,8,9]. These proteins also play a vital role in regulating immune functions [10,11,12], promoting the survival of different immune cells such as monocytes and lymphocytes and influencing cytokine expression [5,10,13].

The synthesis of these proteins takes place in the rough endoplasmic reticulum (RER) of cells in the form of pre- (preNTs) and pro-neurotrophins (proNTs) [14,15]. The yielding of preNTs in the RER gives place to proNTs, which can undergo post-translational changes through the Golgi apparatus to the trans-Golgi network (TGN), leading to the mature form of NTs [14,15]. Some of these post-translational modifications include N-acetylation and carboxyterminal-amidation [15]. These processes in the synthesis of NTs can determine their role and function of the CNS [15]. In addition, proNTs dimerize as homo- or heterodimers [14]. ProNTs can be classified into two groups based on their secretory pathway: the constitutive secretion and the regulated secretion [16]. The first consists of small vesicles that do not need an extracellular stimulus to secrete mature NTs [14], while the regulated secretion consists of more prominent vesicles and requires an extracellular stimulus to secrete them [14]. The secretory pathways are dependent on various factors, such as pH and Ca^2+^ concentration inside of TGN [14].

Furthermore, some studies have demonstrated a relationship between proNTs and brain disorders [3,17], such as brain dementias, strokes, and seizures among others, where the levels of proNGF have been shown to increase after the incident, inducing neuronal apoptosis after the injury [3,18]. Along these lines, proBDNF has been related to facilitating long-term hippocampal depression [3]. Still, the regulation of the signaling pathway of these NTs, which starts with the binding to their corresponding receptors, can induce alterations in the signaling of NTs leading to neurologic defects in the short and long term [17,19,20,21,22]. Due to this, it is essential to highlight the NTs signaling pathways to understand the effect of some factors such as stress, genetics, inflammation, mood, and brain disorders [4,5,17]. It has been described that viral infections can interfere with the normal NTs signaling pathways to increase their infectivity in the CNS or even lead to cognitive impairment and neurobehavioral disorders [23,24,25,26].

In this article, we will mention essential aspects and functions of NT signaling pathways, such as the maintenance of homeostasis and integrity of the CNS. Further, we will discuss the impact of a viral infection in the CNS on the regulation of NTs signaling pathways and the short- and long-term effect on the CNS as a consequence of viral infection.

### Neurotrophins Receptors Signaling Pathways

Two different types of NTs receptors can be found within the CNS cells [14,27]. The Tropomyosin receptor kinase (Trk) binds preferentially to specific NTs to develop different functions in CNS, creating a high-affinity attachment [10,20,28]. These receptors include TrkA, TrkB, and TrkC, where TrkA binds to NGF, TrkB with BDNF and NT-4/5, and TrkC binds to NT-3 (Figure 1) [14,29,30]. Interestingly, it has been described that NT-3 can bind and activate with the other Trk receptors, but with less efficiency than TrkC [31]. Furthermore, it is known that the TrkB receptor is expressed widely in CNS and is independent of the NTs levels, but the activation of the TrkA and TrkC receptors depends on the NT levels in their surroundings [17,28,32].

The lower-affinity receptors are known as 75 kD NT receptors (p75NTR) [28,32]. Although all the NTs mentioned above can bind with this receptor, they do so with less affinity than as seen for the Trk receptors [33]. The p75NTRs are part of the tumor necrosis factor (TNF) family [1,34]. They contain a cysteine and death domain, and each one has a particular feature, participating in the binding to NGF24 or cytotoxic functions, respectively [1,35]. The death domain highlights the similarity between p75NTR receptors and TNF [1,29,32,34]. Differences between p75NTR and TNF are based on the lack of trimerization of p75NTR and the differences in their activation [3,17]. These receptors display more affinity for proNTs than mature NTs [3,17]. It has been described that NTs can modulate robust signaling, binding affinity, and modulation of CNS functions in the presence of p75NTR and Trk receptors, including enhancing cell survival [1,17,19,31]. Notably, the binding of proNTs with p75NTR can induce a signaling cascade with different functions in different cell lines, similar to the mature NTs [36], such as intracellular traffic, protein storage, apoptosis, and synaptic plasticity, among others [3,36,37]. However, activating the p75NTR receptor signaling pathways requires binding to the adaptor proteins and sortilin to bind to proNTs [1,31]. In addition, a balance between proNTs and mature NTs is necessary for the proper development and function of the CNS [3], as was discovered in the case of proBDNF and BDNF in a study with KO proBDNF mice performed by Hua Li et al. [38]. This study found proBDNF to be essential for cell survival, the development of the cells within CNS and the GABAanergic system, promoting motor behavior, and can even cause Huntington’s disease phenotype in mice [17]. However, the study did not show an association between proBDNF levels and BDNF signaling [17]. The lack of proBDNF synthesis may affect the proper signaling, supporting the importance of binding proNTs with p75NTR/Sortilin for the CNS function [37]. As for the case of proNGF, it was described that the increase of proNGF levels could enhance the pro-apoptotic signaling mediated by p75NTR binding, which would lead to an abnormal CNS behavior, which can trigger mood disorders [37].

In the same way, there is an association between mature NGF and proNGF levels in rats exposed to chronic stress based on the increased levels of proNGF and decreased levels of mature NGF in the hypothalamus, but this did not affect the signaling nor the expression of p75NTR and TrkA receptor [17]. Therefore, an indirect relation between levels of proNGF and mature NGF is likely. In addition, while NGF has a regulatory and pro-survival function in the CNS, proNGF promotes a pro-apoptotic function in this tissue [17].

NTs signaling is key to the development of essential functions in the CNS [1,3,17,19,20,31,32,38]. Several signaling pathways are triggered by Trk engagement, but the best characterized are the binding-inducing phosphatidylinositol 3 kinases (PI3K)-Akt, the mitogen-activated protein kinase/extracellular signal-regulated kinase (MAPK-ERK), and the phosphoinositide-specific phospholipase C-γ1 (PLCγ1) [1,20,32,33,39].

The MAPK-ERK pathway is modulated by a signaling cascade that includes Ras, which is involved in neuronal differentiation and cell survival [22,31,33] and leads to the activation of MAPK-ERK signaling [1,20,32]. MAPK-ERK is involved in several functions and events inside the CNS, such as synaptic plasticity and cell survival [19,38]. On the other hand, the signaling via PI3K-Akt can be activated independently or dependently on Ras [33], and in the case of most neuronal cells, the activation of PI3K through Ras induces cell survival signals [33]. Furthermore, it was seen with cells stimulated with NGF, using the NF-kappaB transcription factor [39,40]. Lastly, through its different isozymes, the PLCγ1 signaling pathway regulates various pathways related to neuronal activity and synapse and proper brain development [33].

The signaling triggered by p75NTR receptors contributes to deciding the cell’s fate, promoting either survival or apoptosis [1,19,41]. Cell survival is promoted in the case of the activation of NF-kappaB signaling [19,20,28,41]. However, the activation of p75NTR through ligand biding can also activate the Jun N-terminal kinase (JNKs) signaling cascade [6,17,28,31,41], inducing apoptosis through the expression of Fas-ligand and its receptor [31].

Finally, signaling pathways from endosomes to traffic NTs and their receptors play an essential role in NT signaling [42]. Within this endosomal pathway, the Rab GTPases control several intracellular vesicle traffic, such as the formation of the vesicle and motility, and the lack of these GTPases is associated with neurological disorders and immunodeficiencies [42,43]. The Rab GTPases implicated in the NTs pathway are Rab5 and Rab7, associated with the early endosome and late endosome, respectively [44,45,46]. It has been reported that Rab5 regulates the early entry of the TrkB receptor into the endosomes, allowing the activation of several factors leading to the activation of genes promoted by BDNF in neuronal cells [44]. Due to this, the absence of Rab5 is prejudicial for the correct morphological changes promoted by BDNF, altering the branching process of dendrites [44].

Interestingly, the presence of BDNF has been associated with promoting the movement of Rab5-positive endosomes located in dendrites towards the soma of the neuron [44]. On the other hand, the regulation of the endosomal trafficking of TrkA during neurite projections is under Rab7 [45]. Given that the impairment of Rab7 that causes the accumulation of endosomes with TrkA has been related to neurodegenerative diseases [45], it can be suggested that the Rab7-positive endosomal trafficking of TrkA is a crucial process for the correct development of the CNS.

## 2. Role of Neurotrophin Signaling in CNS Homeostasis

NTs significantly contribute to healthy CNS development and maintenance [47]. Furthermore, most mechanisms mediated by NTs signaling have essential physiological effects which perdure through most animal lifecycles [47,48]. In consequence, alterations in neurotrophic signaling can have substantial ramifications, especially during embryonic development [47,48,49]. This section will discuss the critical physiological roles of NT signaling for CNS homeostasis and its relation to brain diseases.

### 2.1. Neurotrophins in the Brain Development

Due to the extensive effects of NTs in neuronal cells, neurotrophic activity is essential for the proper development and functionality of the brain circuit [50]. For instance, hippocampal and cortical brain structure development is mainly influenced by BDNF and NT-3 signaling [51,52,53]. Additionally, NGF and BDNF during early brain development allow the differentiation of stem cells into neurons and their survival [48]. In general, NTs have been associated with the differentiation of multipotential precursor cells into multipolar and bipolar neurons and oligodendrocytes [48].

The cell fate and behavior depend on the corresponding dominant form of present NTs, due to that NTs display a high affinity for Trk receptors and low affinity for p75NTR; meanwhile, the opposite phenomenon is observed for proNTs [52,53]. Even though these receptors interact directly and can modulate their effects on target cells, p75NTR activation usually leads to apoptosis, while Trk activation promotes cell survival [54]. Both NTs and proNTs activity help regulate the number of neurons surviving, especially in early brain circuit development, where cell survival can be disproportionate, becoming necessary for eliminating some neurons [47].

It has been described that BDNF significantly influences early brain development, promoting stem cell differentiation into neurons and their survival [48]. For instance, in striatum stem cell-derived neuron precursors, the NT loses effect in cell survival after achieving suitable differentiation [49]. This implies that BDNF stimulates neurite outgrowth in striatum neurons but does not act as a survival factor, making it insufficient to prevent death over the time [49]. Moreover, it was shown that mice with impaired production of BDNF during the early stages of brain development experience several synaptic plasticity and transmission problems due to reduced neuron survival after stem cell differentiation [51]. These complications include the impairment of long-term potentiation (LTP) and paired-pulse facilitation (PPF) and the lack of synaptic response to high-frequency stimulation (HFS) [51]. In addition, BDNF insufficiency also causes cognitive alterations in mice and humans, such as elevated aggression, anxiety and depression-like behaviors, and a lack of learning modulation [55,56]. Nevertheless, it has been shown that stimulating the expression of BDNF via physical exercise has a positive impact on the attention span of children with attention deficit hyperactivity disorder (ADHD) [57].

In association with BDNF, the expression of its receptor TrkB has been studied during epilepsy, resulting in observations that it influences the rearrangement of brain neuron circuitry following epileptic seizures [58]. The decrease of this BDNF receptor in neocortical brain areas has been associated with dendritic retraction, but not with cell death following epileptic convulsions [58]. On the other hand, when proBDNF binds and activates its preferred receptor, p75NTR, it has been shown to exhibit pro-apoptotic activity in BDNF target cells [59]. Additionally, BDNF significantly influences the survival of different neuron populations in hippocampal and cortical structures [53,60]. Therefore, abnormally elevated levels of BDNF in the hippocampus and low levels of BDNF in cortical areas have been associated with schizophrenic psychoses [60]. It has also been found that brain-injected BDNF has neuroprotective effects in the ischemic hippocampus of rats [60]. In adults, hippocampal stem cell proliferation and differentiation are regulated by BDNF expression, which is regulated by neurological activity [61]. In turn, neurological activity, along with the expression of BDNF, has been shown to modulate neural plasticity [61,62].

Neurogenesis has been linked to BDNF activity in hippocampal and cortical areas of the macaque monkey embryonic brain and the hippocampus of adult rats [5,63]. This suggests that BDNF plays a role in the proliferation of new neurons through a significant portion of animal lifecycles. BDNF has also aided retinal detachment and reattachment in cats in protecting surviving photoreceptor cells and possibly promoting regenerative responses in peripheral tissues [64].

On the other hand, NGF activity in embryonic CNS tissues has been extendedly documented since the discovery of this NT [23,65]. For instance, the early development of mice’s basal brain cholinergic neuron projections to the hippocampus and cortex is enhanced by NGF [65]. However, the survival of these neurons is independent of NGF exposure [65]. When basal brain cholinergic neurons are developed in the absence of NGF, they can atrophy, which has been associated with neurological disorders such as Alzheimer’s disease [24,66]. In addition, exogenous exposure to NGF on atrophied cholinergic neurons enhances synaptogenesis in cortical tissue [24]. This has sparked many attempts to develop effective NGF treatments for Alzheimer’s disease [25]. Other neurological disorders have been studied concerning NGF activity, such as Huntington’s disease, in which a progressive decrease of hippocampal NGF levels has been observed in different rat model ages [26]. Furthermore, intracerebral injection of this NT in rat models restored spatial working memory and enhanced hippocampal neurogenesis [26,67]. 

Synergistic and combined effects of NGF and BDNF have been documented [23]. In vitro neuronal stem cell differentiation is significantly improved by both NTs combined than NGF or BDNF alone [23]. Combined effects of subtle variations of NGF and BDNF on neuronal cell death, physiological disorders, and cognitive problems have been documented in alcohol-exposed newborns [68,69]. Diverse effects of this NT have been observed in experimental conditions; however, its exact role in unperturbed brain tissues remains elusive [65].

NT-3 activity has shown associative and synergistic effects with other NTs, such as BDNF, in many NT-3 target tissues [70,71]. Such is the case with the extension of the time window for brain plasticity by BDNF and NT-3 exposure on neonatal rat cerebellum [70]. Added effects of BDNF and NT-3 have even made the reprogramming of human dental pulp stem cells into neurogenic and gliogenic neural crest progenitors possible [72]. NT-3 has demonstrated significant influence in aiding the activity of other NTs in multiple target tissues and cells. NT-3 own activity has shown diverse effects and influence over neural circuitry and proliferation [73]. Multiple studies have demonstrated that this NT is not only required for spinal proprioceptive afferent motor neuron connections, but its overexpression has adverse effects on synaptic selectivity between sensory and motor neurons [73,74]. Auditory neuron neurite outgrowth is also positively affected by NT-3 signaling [75]. In contrast to previously described neural proliferative effects, NT-3 can also inhibit cortical neural precursor proliferation via the fibroblast growth factor 2 FGF2 pathway [76]. Furthermore, abnormal NT-3 levels have been associated with neurological disorders, such as autism [51]. In cortical areas, abnormally low levels of NT-3 have been related to schizophrenic psychoses [55]. The effects of NT-3 signaling are diverse and significantly dependent on the biochemical context before tissue exposure, including the presence of other NTs such as BDNF [70].

NT-4/5 has marked effects on brain neuron circuit growth and structure after a proper stem cell differentiation stage in brain development [58,60]. It has been associated with BDNF in significantly influencing neural plasticity, even in a differentiated tissue [57]. For instance, NT-4/5 activity enhances glutamatergic synaptic transmissions in cultured hippocampal neurons [60,77]. Cerebellar granule cells have been positively induced neurite outgrowth when exposed to NT-4/5 [78]. Cultured striatal neuron survival, neurite outgrowth, and biochemical differentiation have also been positively associated with NT-4/5 modulation [79]. Studies in retinal ganglion cells also found positive modulation of neurite outgrowth and cell survival by NT-4/5 together with BDNF [80,81]. These results suggest that NT-4/5 has significant effects on the structure of neural circuitry in multiple brains and non-brain areas [77].

Interestingly, studies have described neuro regenerative and protective effects of NT-4/5 in adult rat neural tissues [82]. For example, the exposure of the axon from the rubrospinal motor neuron to NT-4/5, along with cervical axotomy, could cause cellular regeneration and prevent cell atrophy and death [83]. In addition, neuroinflammation after germinal matrix hemorrhage in basal ganglia is attenuated by NT-4/5-Trkb signaling [82]. These effects have also been found in Parkinson’s disease research in rats, where the efficacy of embryonic nigral grafts has been enhanced by NT-4/5 specifically [84].

Finally, behavioral changes concerning the augmenting serotonin, dopamine, and GABAergic systems are modulated by NT-4/5 expression in the basal ganglia [85]. This NT shows significant effects in multiple neural tissues, including brain areas, such as the hippocampus and striatum, specifically in adult and differentiated tissues, which perdures through most animals’ lifecycles [83]. 

The role of NTs is relevant in a significant portion of animal life and has vast implications for developing healthy brain circuitry [86]. A common factor between most physiological effects of NTs signaling is the promotion of neuron and/or non-neuron, such as glial and epithelial cell survival [87]. However, the homeostasis of the CNS depends not only on NTs promoting cell survival and the development of the brain, but also plays an essential role in modifying the synapsis over time [87]. In the following section, the role of NTs during synaptic plasticity will be discussed.

### 2.2. Synaptic Plasticity

As previously mentioned, NTs play an essential role in maintaining homeostasis in the CNS. Within this maintenance is the regulation of synaptic plasticity, which refers to the capability to strengthen or weaken the synaptic transmission of synapsis [88]. BDNF and NT-3 are the most important and studied NTs contributing to this phenomenon.

The effect of BDNF on synaptic plasticity has been recently studied since it has been identified in specific areas where synaptic plasticity occurs, such as the hippocampus, the cerebellum, and the cerebral cortex [89]. It should be noted that these studies have mainly focused on the hippocampus because it is the region of the brain where synaptic plasticity plays a critical role in learning and memory [90]. As stated above, when mature BDNF is released, proBDNF is also released, which have opposite effects by binding to their respective receptor [90]. Studies with knockout mice for the gene encoding p75NTR have shown that proBDNF can only induce the generation of Long-Term Depression (LTD) upon the binding with p75NTR [91]. Along these lines, other studies have shown a significant decrease in LTP in adult mice models by blocking the expression of BDNF in the brain [89,92]. Therefore, it can be said that mature BDNF binds to TrkB and promotes the generation of LTP in adult mice [89,92]. Interestingly, synaptic plasticity is a phenomenon that is related to aging since the plasticity is reduced as age advances, which is linked to a decrease in BDNF [93]. Nevertheless, this result could be reversed if BDNF is administered exogenously [93].

As previously seen, BDNF impacts synaptic plasticity, which has resulted in studying the role played by the TrkB receptor [94]. This receptor has been observed preferentially located in the sites where neuronal plasticity is produced, carrying out focused signaling [94]. When BDNF interacts with TrKB, different signaling pathways are activated within the neuron [95]. These pathways are characterized by activating transcription factors such as cAMP-response-element-binding Protein (CREB) and CREB-binding protein (CBP), which activate genes that encode proteins involved in the process of neuronal plasticity and cell survival [95].

As previously stated, the expression of BDNF has been associated with neurological disorders [3,17]. For example, in patients with Alzheimer’s Disease, it has been observed that BDNF expression decreases drastically compared to control subjects of the same age [96]. Additionally, it was found that the protein levels that form TrkB also decrease in these patients, and the CREB signaling is altered by the presence of the β-amyloid peptide, which is one of the main peptides associated with AD [96,97]. These results suggest that the disease, through the use of β-amyloid peptide, blocks the correct signaling of CREB, affecting the plasticity in these patients [96,97]. Another disease where the role of BDNF has been studied is schizophrenia, which is characterized by cognitive impairment [98]. It has been reported that patients who have schizophrenia had serum levels of BDNF significantly lower than controls [99]. Additionally, a correlation was made between serum BDNF levels and a cognitive performance test score that showed a positive association, suggesting that low BDNF expression plays a determining role in the development of the disease [99].

In contrast to BDNF, NGF has been less studied than other NTs, but not because it is less important. It has been seen that the metabolism of NGF in AD is altered, meaning that cholinergic neurons cannot have optimal growth and plasticity since they depend on this NT for complete development [100].

In a study in adult rats, NGF levels were modified and the effects of NGF on hippocampal neurons, LTP, and learning were examined [101]. It was observed that the increase in NGF produced a significant increase in markers of the formation of new neural networks [101]. On the other hand, a blockade in releasing this NT significantly reduced LTP, impairing spatial memory in rats [101]. Thanks to these results, the researchers concluded that NGF plays an essential role in regulating mechanisms related to plasticity and memory, but the mechanisms in which NGF may intervene are still being studied.

Lastly, NT-3 plays an essential role in synaptic plasticity, which is highly expressed in the dentate gyrus of the hippocampus [102]. In a study using knockout mice for the gene that codes for NT-3, it was observed that NT-3 could regulate the differentiation of neuronal cells in the dentate gyrus, being very important for neurogenesis to occur and thus contributing to plasticity synapse in this region of the hippocampus [102]. 

A study investigated the role of NT-3 in hippocampal plasticity and memory in mouse models [102]. They observed that a blockade of the release of NT-3 generated a deterioration in the LTP, this being observed in neuronal synapses [102]. On the other hand, NT-3 mutant mice showed deficits in spatial memory tests [102]. These results indicate that NT-3 is just as crucial as other NTs in synaptic plasticity, participating in mechanisms still being studied.

### 2.3. Role of Neurotrophins in Oligodendrocytes Development and Myelinization

It is known that NTs have an essential role in the myelination process, which is critical for the conduction of the nerve impulse [103,104,105]. A study performed on BDNF-/- mice showed fewer oligodendrocytes and low levels of the myelin basic protein (MBP) in different brain areas of wild-type mice [106]. These data showed a direct relationship between the expression of BDNF and MBP, but it is not the only factor involved in oligodendrocytes maturation because MBP expression was not absent in the brain of BDNF-/- mice, indicating that it is produced by another pathway [106,107]. Indeed, it has been described that BDNF has effects on the proliferation of oligodendrocytes progenitor cells (OPCs) and promotes its differentiation through its interaction with TrkB and the subsequent signaling pathway (MAPK/PI3K), described earlier [108,109]. The final effect of BDNF on the oligodendrocytes is the up-regulation of MBP, which was evaluated in BDNF+/− mice that present myelination deficit in the optical nerve, brain, and spinal cord during the postnatal development [110]. In demyelinating and inflammatory diseases such as Multiple Sclerosis (MS), it has been observed that BDNF levels were elevated in brains with an inflammatory lesion, where the primary source of this NT was the immune cells [111,112]. Moreover, low plasma levels of BDNF were observed in MS patients compared to controls and after relapse [113,114]. According to these findings, BDNF has a crucial role in MS [115].

On the other hand, NGF has the opposite effects on Schwann cells and oligodendrocytes [116]. Experiments performed in cell cultures showed that NGF can promote the myelinization of dorsal ganglia roots (DGRs) neurons by Schwann cells [116]. Contrary to this, OPCs cultured onto DGRs and in the presence of NGF showed a decreased myelinization, a lower number of differentiated oligodendrocytes, and inhibition of the maturation of oligodendrocytes [116]. These effects are through the TrkA, which is present in the DGRs [116]. Meanwhile, the inhibition of the differentiation of the OPCs is through its interaction with p75NTR [117]. The neutralization of NGF can revert the impairment in myelinization, demonstrating the negative regulation of oligodendrocytes [118]. In MS patients, it has been described that NGF is elevated in cerebrospinal fluid (CSF). In lesions zones of the brain, immature and apoptotic oligodendrocytes can be found expressing high levels of p75NTR [119,120].

NT-3 has been shown to promote OPCs proliferation, differentiation, and survival [121,122,123]. Also, NT-3 overexpression can promote and induce oligodendrogenesis in the injured spinal cord, besides the myelinization of ingrowing axons [105]. Moreover, NT-3, together with BDNF, promotes axonal survival after spinal cord injury preventing neuronal damage and apoptosis [114,124]. However, concerning MS, the role of NT-3 is poorly studied.

## 3. Effect of CNS Viral Infections on Neurotrophin Signaling

As was mentioned earlier, NTs signaling pathways are essential for several processes, including synaptic plasticity [61,62]. Therefore, during CNS damage or pathologies, such as traumatic brain injury, neurodegenerative diseases, or viral infection, changes in the levels of the NTs and/or their receptors can cause neuropsychiatric disorders and cognitive impairment [5,63,64]. Significantly, viral infections can promote alterations in the NT levels and use their receptors for infection of the target cells. This section will discuss how neurotropic viral infection can modulate the NTs signaling pathways.

### 3.1. Findings and Effects during Viral Infection

Due to the critical role of NTs in brain function, several reports have shown that the levels of these molecules in the CSF are a helpful indicator of brain damage, mainly in post-traumatic injury and cognitive impairment [125,126]. Accordingly, the NTs also can be detected during a neurotropic viral infection or in neurocognitive disorders provoked by the virus [24,25,26]. Among the viral agents that can alter the NTs signaling, human immunodeficiency virus type 1 (HIV-1) is the one that has been more studied [26]. Still, other viruses can interfere with the signaling of NTs, including Epstein Barr virus (EBV), herpes simplex virus-1 (HSV-1), influenza virus, human respiratory syncytial virus (hRSV), and severe acute respiratory syndrome coronavirus 2 (SARS-CoV-2) [67,127,128].

The HIV-1 infection is known for causing an ineffective immune response and can provoke mild or severe cognitive impairments, known as HIV-associated neurocognitive disorder (HAND) [67,69]. In this context, NTs such as BDNF have been observed in the CSF from patients with HAND [25,68]. According to this, lower levels of BDNF were found on CSF from patients with mild HAND or HIV-associated dementia (HAD) than in patients without neurocognitive symptoms [68]. Furthermore, another study with patients with HAD showed decreased BDNF and NT-3 in CSF, which correlates negatively with a poor neurological score [25,129]. These data suggest that a deleterious effect of HAND or HAD may be due to the loss of the neuroprotection given by the NTs.

Studies regarding specific effects of the virus, or its proteins, on the NT signaling pathway are crucial to understanding how the virus affects it. In vitro studies using microglial cell lines have shown that stimulation with HIV-1 gp120 proteins, a glycoprotein important in the pathogenesis developed by HIV, can promote the increment of both proBDNF and mature BDNF (Figure 2) [130]. Additionally, the increment of BDNF was associated with the activation of the Wnt/β-catenin signaling pathway due to stimulation with HIV-1 gp120 protein [130]. The increment of BDNF levels after the stimulation with HIV-1 gp120 protein could be explained as a protective mechanism against the infection. Other HIV-1 proteins can alter normal NTs levels, such as transactivator of transcription (Tat) [131]. By using primary neuronal cultures, it was possible to elucidate that the HIV-1 Tat protein could induce neuronal impairment due to the neurotoxicity and downregulation of the CREB pathway, decreasing the levels of BDNF [131]. The stimulation of neuronal cells with HIV-1 Tat protein interferes with the MAPK/ERK pathway that NGF induces, and as a result, this protein blocks the signaling for neuronal survival [132]. Interestingly, it has been reported that Rab5 and Rab7 are involved in the first steps of the infection of astrocytes with HIV-1, and the inhibition of these GTPases prevents the infection of this type of glial cells [133]. Based on this, it can be suggested that HIV-1 induces the endosomal trafficking of TrkA and TrkB to infect astrocytes [44,45]. However, not only can viral proteins affect NTs, but the other way around is possible as well, as shown next. In vitro studies have shown that NGF can stimulate viral replication, promoting the activation of the HIV-1 gene transcription [134]. Furthermore, anti-NGF neutralizing antibodies reduce HIV-1 production in macrophages, promoting macrophage survival and preventing the pathogenic events caused by the infected macrophages [134,135,136]. However, besides their effects on HIV-1 replication, both NGF and BDNF can recover neuronal plasticity after HIV-1 infection [134]. Despite being the most studied viral agent associated with NTs, there is still much to know regarding their interaction.

HSV-1 is known for causing herpetic gingivostomatitis when symptomatic and can remain in a latency state in peripheral neurons and reactivate intermittently to replicate [128,137]. However, the mechanisms responsible for this are still unknown [137]. In an in vitro study using a primary neuron cell culture, it has been found that the latency state of this virus can be maintained thanks to the constant signaling through the PI3-K pathway, which, as it was described previously, is triggered by the binding of NGF to TrkA [137]. Furthermore, it has been studied that treatment of this primary neuron cell culture with anti-NGF antibodies promotes HSV-1 replication [137]. Therefore, these results suggest that NGF, through its signaling pathway, can promote the latency state of this virus. Interestingly, the GTPase Rab5 is necessary for the endocytosis of the glycoproteins from HSV-1 and for enveloping [138]. Therefore, it can be suggested that the Rab5-positive endosome trafficking due to the infection with HSV-1 could be promoting the BDNF/TrkB signaling [44].

Another virus belonging to the Herpesviridae is the EBV, which is known for causing Hodgkin’s disease, among other lymphomas [127]. Interestingly, as seen with HSV-1, the PI3-K signaling pathway also controls the replication cycle of EBV [137]. Contrary to the observed in HAND or HAD patient samples, increased levels of BDNF and NGF were found in CSF from an infant with meningoencephalitis (ME) induced by EBV [25]. Additionally, a correlation was observed between the NTs overexpression with the amounts of lymphocytes in CSF and the disease severity, suggesting a role in the inflammation during the EBV infection [25]. However, little is known regarding the role of NTs during this viral infection with the herpes virus in patients, and more studies are needed.

Respiratory viruses have also been studied regarding their effects on the levels of NTs. For example, the influenza virus is a respiratory virus known for causing pathologies related to the respiratory tract [139]. It has been reported that patients positive for the Influenza virus present a significant increment in the levels of BDNF and NGF compared to the controls [140,141]. In addition, higher BDNF and NGF levels were detected in patients with severe symptoms than in patients with mild symptoms [140,141]. This last result presents a possible marker for the severity of the disease due to infection with the influenza virus [140,141].

Interestingly, in vivo studies have demonstrated that the levels of BDNF decrease in the hippocampus of influenza virus-infected mice, regardless of whether they are neurotropic (H3N2 and H7N7) or non-neurotropic (H1N1) [142,143]. This suggests that the influenza virus’s effect on BDNF can be accomplished without necessarily being in the CNS. However, it has only been reported that the levels of NT-3 decrease in the presence of neurotropic strains of influenza virus, while NGF decreases in the presence of a non-neurotropic strain of influenza virus [142,143]. In vitro studies using HeLa cells demonstrated that the absence of Rab5 or Rab7 significantly decreases infection with influenza virus, reducing the traffic of this virus [144]. It would be interesting to evaluate this phenomenon in neurons since the infection with influenza virus could promote the endosomal trafficking of both TrkA and TrkB and, as a result, induce these signaling pathways [44,45].

The hRSV is the principal respiratory virus responsible for causing infections in the lower respiratory tract in infants [145]. It has been detected in infants positive for hRSV infection a significant increment of BDNF and NGF levels compared to the controls [146]. Even though it has not been reported, the differences in the concentrations of both NTs could help predict the severity of the disease, as in the case of the influenza virus. In addition, in vivo studies have demonstrated that neutralizing NGF decreased lung inflammation in rats infected with hRSV, protecting against this infection [147]. This effect was possible because NGF promotes a Th2 type of response instead of a Th1 during the infection [147].

Interestingly, it has been seen that the infection with hRSV in bronchial cells promotes the increment of NGF levels and TrkA, which is the receptor that has a high affinity for NGF, while inducing a decrease of p75NTR, which has a low affinity for NGF [148]. Since hRSV can also infect cells within the CNS [149], it could be suggested that the virus can produce this effect to promote the infection within the CNS. However, more studies are needed to elucidate how the infection with this virus affects the NTs within the CNS.

SARS-CoV-2 is a novel respiratory virus known for affecting the respiratory tract [150]. It has been reported that the patients positive for SARS-CoV-2 present increased levels of BDNF compared to controls, and a high BDNF level was associated with the worst prognosis for the patient [151,152]. Additionally, infection with SARS-CoV-2 induces the β-NGF/TrkA signaling pathway in lymphocytes [153]. However, even though there are no studies associating the effect of the infection with NTs within the CNS, it does not mean that there is no relation. Therefore, the effect on the CNS remains to be elucidated.

### 3.2. Long-Term Effect

Neurotropic viruses can interfere with the correct signaling pathway of NTs and cause neurological pathologies. However, these pathologies can persist in the long term, causing disorders such as learning and memory impairment, mood and speech disorders, and degenerative diseases [154].

The infection with HIV can cause cognitive problems, such as HAND or HAD. However, learning and memory impairment cases are detected in HIV-positive patients that can be associated with a decrease in BDNF levels (Figure 3). In these patients, the highest concentration of the virus resides within the hippocampal region of the whole brain, which is the region related to learning and memory [155,156]. Due to this, and because the learning and memory process has been associated with BDNF signaling, it was studied whether the administration of additional BDNF could restore this impairment [156]. It was seen that the administration of external BDNF in mice with HIV could increase the synaptogenesis in the hippocampal region, which is a process related to the mechanism of developing learning and memory processes [156,157]. Additionally, it improved both learning and memory impairment after a week of administrating BDNF to the mice [156]. These results show a positive correlation between BDNF and the increase in synaptogenesis in the hippocampal region and, as a result, learning and memory improvement [156]. Interestingly, during a study performed with patients with HIV, it was possible to establish an association between reduced BDNF levels and depression, struggle concentrating, and memory loss [158]. These studies demonstrate a positive association between BDNF levels and improving cognitive impairment.

In the case of HSV-1, few studies have been performed regarding the long-term sequels after affecting NT signaling. Interestingly, it has been reported that an increase in the number of HSV-1 viral copies can decrease the BDNF levels [159]. Together with a reduction in NMDA receptors, this phenomenon induces problems in synaptic plasticity and the accumulation of Aβ protein that can lead to Alzheimer’s disease [159]. Even though there is no information regarding the association between the infection with EBV and its effect in the long term, it can be suggested that EBV can promote similar phenomena since they both belong to the same family.

Respiratory viruses can also interfere with the signaling of NTs and cause long-term sequels. Even though no specific studies associate the interference of the NTs signaling due to the infection with respiratory viruses can cause a long-term sequel, it is possible to suggest an association. For example, the influenza virus is one of the respiratory viruses known for causing long-term consequences, such as learning and memory impairment [154]. It has been demonstrated that infection with the influenza virus could decrease the levels of BDNF and NGF in the hippocampal region [142]. This may contribute to the long-term effects of influenza in the brain, such as cognitive impairment [142,154].

Regarding hRSV and SARS-CoV-2, learning and memory impairment cause cognitive problems that have been previously described [149]. In this line, it could be suggested that, as previously mentioned in the case of the influenza virus [142,154], the learning and memory impairment could be associated with an alteration in the levels of BDNF. Interestingly, after the infection with SARS-CoV-2, the patients that developed symptoms such as depression and anxiety presented lower levels of BDNF compared to the controls [160]. However, this needs to be further evaluated.

## 4. Conclusions

NTs are proteins involved in several essential functions of the CNS and other systems, including the immune and respiratory [1,7]. The activation of the signaling pathways of these NTs, such as NGF, BDNF, NT-3, NT-4, and NT-5, start with the recognition of these NTs by cellular receptors, such as Trk and p75NTR [27,33]. The correct functioning of these signaling pathways contributes to maintaining the homeostasis of the CNS through proper brain development, neuronal survival, and plastic synapsis [47]. However, the role of these NTs and pro-NTs can be modified by CNS injuries and the presence of several viruses [67,127,128]. Some viruses, such as HIV and HSV, have been studied and how alter the levels of NTs and the effects that these viruses have on the CNS, both in short and in the long term, are known. Other viruses, such as Influenza virus, hRSV, and SARS-CoV-2, have not been studied regarding the relationship between the consequence in the CNS and the interference with the NTs. Nevertheless, due to the consequences reported by these respiratory viruses, it is possible to suggest that the infection can alter the levels of NTs and, therefore, could cause cognitive impairment in both the short- and the long-term.

As seen in this work, the NTs are necessary not only for the homeostasis of the CNS; therefore, more studies are needed to understand how viruses can modify their function and cause several effects in patients.

## Figures and Tables

**Figure 1 ijms-23-05817-f001:**
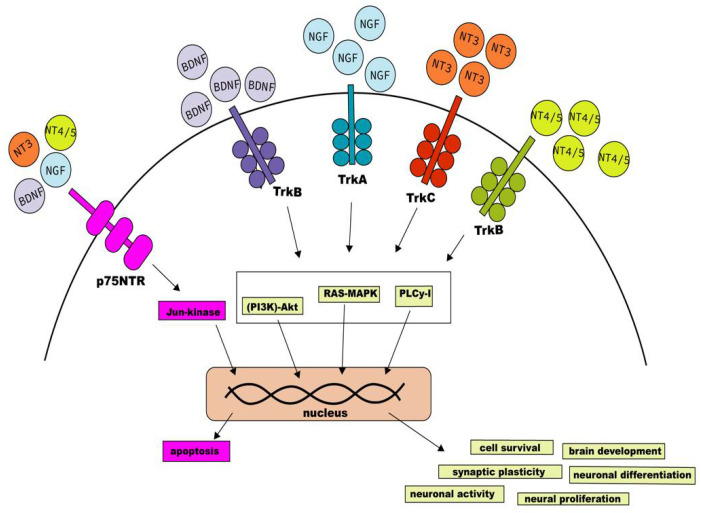
Neurotrophin signaling pathways. The figure shows different NTs (NGF, BDNF, NT3, and NT4/5) in a distinctive color with their high-affinity receptors (TrkA, TrkB, and TrkC) and the p75NTR, which can bind to all the NTs mentioned above. The figure also shows the main intracellular signaling pathways and the different effects on the cell [14,27,29].

**Figure 2 ijms-23-05817-f002:**
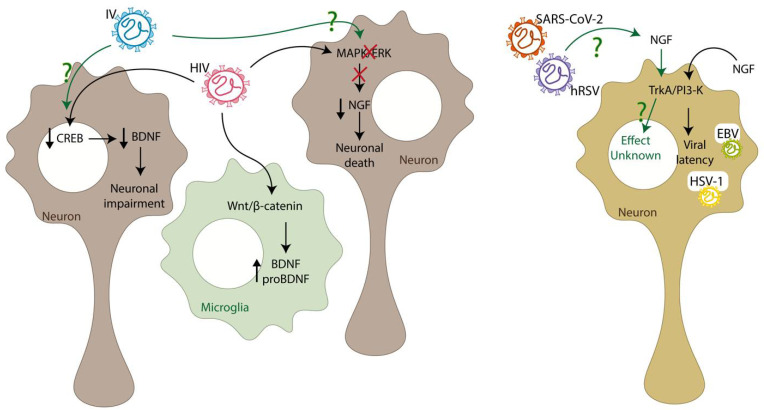
Viruses interfere with the normal signaling pathways of neurotrophins. The human immunodeficiency virus (HIV) can promote the increment of both proBDNF and mature BDNF in microglial cells. Additionally, HIV can downregulate the CREB pathway, decreasing the levels of BDNF and causing neuronal impairment. Finally, HIV can interfere with the MAPK/ERK pathway, decreasing the NGF levels and inducing neuronal death. The influenza virus (IV) lowers both BDNF and NGF levels. However, the pathway that alters to cause this effect has not been described. Regarding Epstein Barr virus (EBV) and herpes simplex virus-1 (HSV-1), their viral latency is induced by activating the PI3-K pathway activated by the NGF/TrkA association. Lastly, in the case of human respiratory syncytial virus (hRSV) and severe acute respiratory syndrome coronavirus 2 (SARS-CoV-2), it is known that they increase the levels of NGF and its high-affinity receptor TrkA. However, the effect that this interaction has remains unknown.

**Figure 3 ijms-23-05817-f003:**
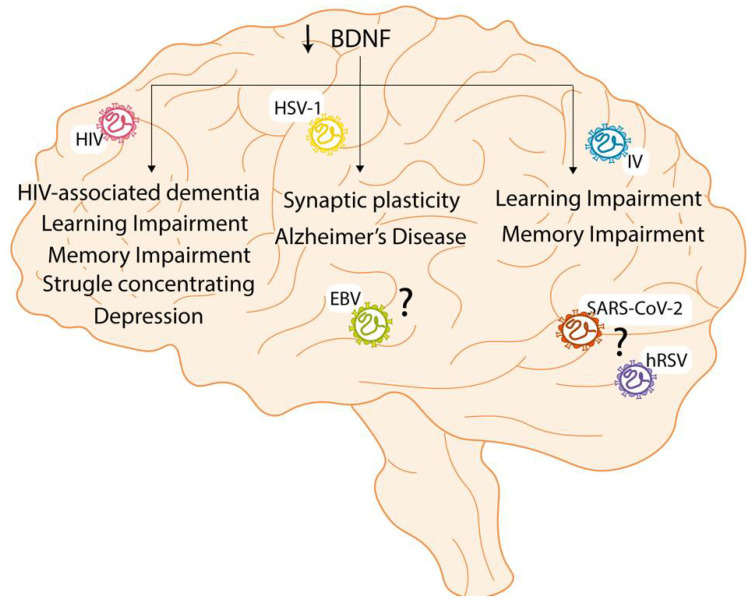
Neurological sequels in the long term due to the viral infection decreasing BDNF levels. The human immunodeficiency virus (HIV) can decrease BDNF levels and cause HIV-associated dementia (HAD), learning and memory impairment, struggle concentrating, and depression. The influenza virus (IV) can decrease BDNF levels and cause learning and memory impairment. In the case of human respiratory syncytial virus (hRSV) and severe acute respiratory syndrome, coronavirus 2 (SARS-CoV-2) are both respiratory viruses associated with learning and memory impairment, suggesting that the decrease in the levels of BDNF causes these impairments. Lastly, herpes simplex virus-1 (HSV-1) can decrease BDNF levels and cause synaptic plasticity and Alzheimer’s disease. Because Epstein Barr virus (EBV) belongs to the Herpesviridae, it can be suggested that it might decrease BDNF levels and cause these pathologies.

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
