# Peer review of "Neurotrophin Signaling Impairment by Viral Infections in the Central Nervous System"

_ijms, 2022, doi:10.3390/ijms23105817_

Round 1

Reviewer 1 Report

Please, see the attached PDF.

Author Response

Answer to Reviewer 1

1.- Reviewer 1: The manuscript “Neurotrophin signaling impairment by viral infections can lead to neurological alterations” examines the importance of neurotrophins in the central nervous system and their signaling pathways, and it is focused in their relationship with viral infections, reviewing how viruses can affect neurotrophins and vice versa. It is sound and interesting for the field and it is well organized and described.

Answer: We thank the reviewer for the comments made to this article.

2.- Reviewer 1: However, under my point of view, major revision of the manuscript should be perform to be acceptable. First, the role of neurotrophins in myelination should be tackled since it is known that neurotrophins are involved in that process and in demyelinating diseases in the central nervous system such as Multiple Sclerosis. Indeed, neurotrophins are involved in migration, maturation and myelination capacity of oligodendrocytic cells.

Answer: As requested by the Reviewer, the manuscript was modified to describe the contribution of neurotrophins to the myelination process (Page 8, Lines 353-388).

3.- Reviewer 1: Second, viruses may use a common transport pathway involved in neurotrophin signal transduction pathways in which neurotrophins and their receptors traffic via Rab5- and 7-positive vesicles. This should be mentioned.

Answer: As requested by the Reviewer, the manuscript was modified to include the endosomal trafficking of neurotrophins via Rab5 and Rab7 (Page 4, Lines 142-159) and to . discuss as to how viruses use this pathway (Page 9, Lines 429-432; Page 10, Lines 461-465; Page 11, Lines 489-493).

4.- Reviewer 1: Besides reference 1, another more recent also interesting references could be added, for instance “Neurotrophin regulation of neural circuit development and function”, by Hyungju Park and Mu-ming Poo, or “Neurotrophins and Proneurotrophins: Focus on Synaptic Activity and Plasticity in the Brain”, by Julien Gibon and Philip A Barker

Answer: As requested by the Reviewer, we have added the suggested references (Page 1, Line 32).

5.- Reviewer 1: Besides reference 5, other references could also be relevant, such as “Interactions between the cells of the immune and nervous system: neurotrophins as neuroprotection mediators in CNS injury” by Tabakman et al or “Neuronal influence behind the central nervous system regulation of the immune cells” by Chavarría and Cárdenas.

Answer: As requested by the Reviewer, we have added the suggested references (Page 1, Line 36).

6.- Reviewer 1: LINE 53. The sentence “some studies have demonstrated a relationship between proNTs and brain disorders [13,14], such as brain injuries” is not accurate. Brain injury is not a type of brain disorder. Brain disorders are diseases such as dementias, Alzheimer, Parkinson, stroke, seizure, etc. whereas “brain injury” has a broader meaning: brain damage caused by a traumatic or non-traumatic event. For instance, stroke (a brain disorder) may cause brain injury. The whole sentence should be rewritten.

Answer: As requested by the Reviewer, we have rewritten the sentence (Page 2, Line 54).

7.- Reviewer 1: LINE 61. “It has been described that viral infections can interfere with the normal NTs signaling pathways to increase their infectivity in the CNS or even lead to cognitive impairment and neurobehavioral disorders [20–25]”.Reference 20 is not related to this statement, it describes the associations of the BDNF Val66Met and 5-hydroxytryptamine transporter linked promoter region (5-HTTLPR) polymorphisms with cognitive function in elderly Korean individuals, but it has nothing to do with viral infections.

Answer: As requested by the Reviewer, we have modified the reference (Page 2, Line 64).

8.- Reviewer 1: LINE 61. References 21 is not either related to that statement, that paper analyzes whether the concentration of IL-1β and IL-6 and neurotrophins (nerve growth factor (NGF), brain-derived neurotrophic factor (BDNF), glial-derived neurotrophic factor (GDNF)) in the cerebrospinal fluid (CSF) of children with traumatic brain injury correlates with the severity of the injury and its neurologic outcome.

Answer: As requested by the Reviewer, we have modified the reference (Page 1, Line 32).

9.- Reviewer 1: LINE 61. References 22 is not either related to that statement, that paper evaluates BDNF as a biomarker for post-traumatic brain injury depression, cognitive impairment, and functional cognition in a prospective cohort with severe traumatic brain injury.

Answer: As requested by the Reviewer, we have modified the reference (Page 1, Line 36).

11.- Reviewer 1: LINE 61. It could be interesting to add here that viruses may use a common transport pathway involved in neurotrophin signal transduction in which neurotrophins and their receptors traffic via Rab5- and Rab7-positive vesicles (“Virus Infections in the Nervous System” Koyuncu et al 2015). If you “discuss the impact of a viral infection in the CNS on the regulation of NTs signaling pathways” (LINE 66), then it is relevant to include that relationship.

Answer: As requested by the Reviewer, the manuscript was modified to include the endosomal trafficking of neurotrophins via Rab5 and Rab7 (Page 4, Lines 142-159; Page 9, Lines 429-432; Page 10, Lines 461-465; Page 11, Lines 489-493).

12.- Reviewer 1: LINE 74. “TrkA bind to NGF, TrkB with BDNF and NT-4/5, and TrkC 76 binds to NT-3 [10,28].” Here it would be appropriate to include other relevant references such as “The neurotrophic factors brain-derived neurotrophic factor and neurotrophin-3 are ligands for the trkB tyrosine kinase receptor” (Soppet el al. Cell 1991)

Answer: As requested by the Reviewer, we have added the suggested reference (Page 2, Line 77).

13.- Reviewer 1: LINE 335. “during a CNS pathology such as brain injury…” Brain injury is not exactly a pathology, it is a too wide term; it means harm to the brain caused by trauma, stroke, infection, neurodegenerative diseases, etc. Perhaps you mean traumatic brain injury?

Answer: As requested by the Reviewer, we have modified the sentence (Page 9, Lines 389-390).

14.- Reviewer 1: LINE 347, 383, 414, 474 and elsewhere. Regarding the nomenclature of viruses, ICTV recommends that a virus name should not be italicized nor written with a capital, (unless they are proper nouns): human immunodeficiency virus type 1 (HIV-1) herpes simplex virus type 1 (HSV-1), influenza virus, severe acute respiratory syndrome coronavirus 2… etc. Only species names are written in italics with the first word with a capital letter: “the species Herpes simplex type 1…”

Answer: As requested by the Reviewer, we have corrected the viruses name (Page 9, Lines 400-404; Page 10, Lines 441,445,447,449); Page 11, Lines 476, 484, 488,489,498; Page 12, Lines 540,541,543,545)

15.- Reviewer 1: LINE 403. “the herpes virus family”. Please, use herpesvirus (or Herpesviridae)

Answer: As requested by the Reviewer, we have modified the reference (Page 11, Line 465; Page 12, Line 547).

16.- Reviewer 1: LINE 394. Please, avoid “the “ HSV-1, use instead just “HSV-1”, without the article

Answer: As requested by the Reviewer, we have deleted the word “the” from the sentence (Page 10, Line 452).

17.- Reviewer 1: LINE 443. “SARS-CoV-2 is a novel respiratory virus known for affecting the respiratory tract [111]” That reference is focused in immune response and vaccines, another more general paper would be more appropriate here.

Answer: As requested by the Reviewer, we have changed the reference (Page 11, Line 509).

18.- Reviewer 1: LINE 467. “These results show a positive correlation between the increase in synaptogenesis in the hippocampal region and learning and memory improvement”. Do you mean that “These results show a positive correlation between BDNF and the increase in synaptogenesis in the hippocampal region and, as a result, learning and memory improvement”?

Answer: As requested by the Reviewer, we have corrected the sentence (Page 12, Lines 532-533).

19.- Reviewer 1: The role of NTs in myelination has not been tackled. It is known that BDNF enhances CNS myelination during development and is neuroprotective after demyelination (“Neuroprotection on Multiple Sclerosis: A BDNF Perspective”. Xiao 2012). The role of NTs in myelination/demyelination and Multiple Sclerosis should be tackled and additional references should be included, for instance a. “Effects of Neurotrophic Factors in Glial Cells in the Central Nervous System: Expression and Properties in Neurodegeneration and Injury” by Pöyhönen et al. Frontiers in Physiology 2019 b. “The Role of Neurotrophins in Multiple Sclerosis—Pathological and Clinical Implications” by Kalinowska-Lyszczarz et al. IJMS 2012 c. “Brain-Derived Neurotrophic Factor in Central Nervous System Myelination: A New Mechanism to Promote Myelin Plasticity and Repair” by Fletcher et al. IJMS 2018

Answer: As requested by the Reviewer, we have included a new sub-section to address this point (Pages 8-9, Lines 351-385).

We would like to thank the Reviewers and the Editors for their time and effort in handling this manuscript and hope that the current revised manuscript is acceptable for publication in International Journal of Molecular Sciences.

Reviewer 2 Report

Minor revision

Section 2.1. Neurotrophins in the brain development

Line 207: Reference 23 does not apply to Alzheimer's disease. Correct.

Section 3.1. Findings and effects during viral infection

Please correctly cite the references in this section

Line 346: is [22–25], it should be [22, 24, 25], without 23

Line 348: Incorrect citation of the reference, to change 23 to one related to the topic. I think the Authors meant 25.

Line 351: delete 23

Line 406-408: jest „Contrary to the observed in HAND or HAD patient samples, increased levels of BDNF and NGF were found in CSF from an infant with meningoen-  cephalitis (ME) induced by EBV [23]”. How do the Authors know that "induced by EBV". There is no such information in 23? A must improve.

Line 406-412: Mistake in the reference. Why 23?

Line 447: is NGF/TrkA; it should be β-NGF/TrkA

Section 4. Conclusions

Line 515: is [23,58,94,95]. delete 23.

Author Response

Answer to Reviewer 2

1.- Reviewer 2: Line 207: Reference 23 does not apply to Alzheimer's disease. Correct.

Answer: As requested by the Reviewer, we have modified the reference (Page 6, Line 222).

2.- Reviewer 2: Line 346: is [22–25], it should be [22, 24, 25], without 23

Answer: As was requested by the Reviewer, we have deleted the reference (Page 9, Line 399).

3.- Reviewer 2: Line 348: Incorrect citation of the reference, to change 23 to one related to the topic. I think the Authors meant 25.

Answer: As requested by the Reviewer, we have modified the reference (Page 9, Line 401).

4.- Reviewer 2: Line 351: delete 23

Answer: As requested by the Reviewer, we have deleted the reference (Page 9, Lines 404).

5.- Reviewer 2: Line 406-408: jest „Contrary to the observed in HAND or HAD patient samples, increased levels of BDNF and NGF were found in CSF from an infant with meningoencephalitis (ME) induced by EBV [23]”. How do the Authors know that "induced by EBV". There is no such information in 23? A must improve.

Answer: As was requested by the Reviewer, we have corrected the reference (Page 11, Line 469).

6.- Reviewer 2: Line 406-412: Mistake in the reference. Why 23?

Answer: As requested by the Reviewer, we have corrected the reference (Page 11, Line 472).

7.- Reviewer 2: Line 447: is NGF/TrkA; it should be β-NGF/TrkA

Answer: As requested by the Reviewer, corrected the sentence (Page 12, Line 512).

8.- Reviewer 2: Line 515: is [23,58,94,95]. delete 23.

Answer: As requested by the Reviewer, we have deleted the reference (Page 12, Line 581).

We would like to thank the Reviewers and the Editors for their time and effort in handling this manuscript and hope that the current revised manuscript is acceptable for publication in International Journal of Molecular Sciences.

Reviewer 3 Report

This is an interesting and timely Review.

Before the manuscript is accepted, the authors are kindly request to address the following two issues:

i) The Introduction should be shortened.

At present,  lines 30-332 (for a total of 303) cover aspects, although of interest and of relevance, which are not strictly related to main topic "Neurotrophin signaling impairment by viral infections".

Indeed only a minor part of the Introduction encompassing lines 334-507 (for a total of 174)  is dedicated to the main topic.

ii) Early published contributions addressing the main topic have not been discussed or cited.  The authors are kindly requested to include for example:

Garaci E. et al. doi:10.1073/pnas.1332627100

Garaci E. et al. doi:10.1073/pnas.96.24.14013

Piedimonte G. et al. doi:10.1152/ajplung.00365.2011

Hondermarck H. et al. doi:10.1096/fba.2020-00015

Duarte J.G. et al. doi:10.1007/s00436-018-5838-2

Senecal V. et al. doi:10.1002/glia.23904

Author Response

Answer to Reviewer 3

1.- Reviewer 3: The Introduction should be shortened. At present, lines 30-332 (for a total of 303) cover aspects, although of interest and of relevance, which are not strictly related to main topic "Neurotrophin signaling impairment by viral infections". Indeed only a minor part of the Introduction encompassing lines 334-507 (for a total of 174) is dedicated to the main topic.

Answer: As requested by the Reviewer, we have changed the title to point out more clearly that the focus of the article is the signaling of neurotrophins within the CNS and how viral infections can interfere with them and cause neurological alterations.

2.- Reviewer: Early published contributions addressing the main topic have not been discussed or cited.  The authors are kindly requested to include for example: Garaci E. et al. doi:10.1073/pnas.1332627100; Garaci E. et al. doi:10.1073/pnas.96.24.14013; Piedimonte G. et al. doi:10.1152/ajplung.00365.2011; Hondermarck H. et al. doi:10.1096/fba.2020-00015; Duarte J.G. et al. doi:10.1007/s00436-018-5838-2; Senecal V. et al. doi:10.1002/glia.23904

Answer: As requested by the Reviewer, we have mentioned and included some references to the example (Page 12, Lines 435-436).

We would like to thank the Reviewers and the Editors for their time and effort in handling this manuscript and hope that the current revised manuscript is acceptable for publication in International Journal of Molecular Sciences.

Round 2

Reviewer 1 Report

I consider that the manuscript can be accepted in its present reviewed form.